# Beat Detection Recruits the Visual Cortex in Early Blind Subjects

**DOI:** 10.3390/life11040296

**Published:** 2021-03-31

**Authors:** Rodrigo Araneda, Sandra Silva Moura, Laurence Dricot, Anne G. De Volder

**Affiliations:** 1Motor Skill Learning and Intensive Neurorehabilitation Laboratory (MSL-IN), Institute of Neuroscience (IoNS; COSY Section), Université Catholique de Louvain, 1200 Brussels, Belgium; rodrigo.araneda@uclouvain.be (R.A.); sandra.silvamoura@student.uclouvain.be (S.S.M.); 2Institute of Neuroscience (IoNS; NEUR Section), Université Catholique de Louvain, 1200 Brussels, Belgium; laurence.dricot@uclouvain.be

**Keywords:** beat perception, rhythm perception, multisensory integration, touch, vision, early blindness

## Abstract

Using functional magnetic resonance imaging, here we monitored the brain activity in 12 early blind subjects and 12 blindfolded control subjects, matched for age, gender and musical experience, during a beat detection task. Subjects were required to discriminate regular (“beat”) from irregular (“no beat”) rhythmic sequences composed of sounds or vibrotactile stimulations. In both sensory modalities, the brain activity differences between the two groups involved heteromodal brain regions including parietal and frontal cortical areas and occipital brain areas, that were recruited in the early blind group only. Accordingly, early blindness induced brain plasticity changes in the cerebral pathways involved in rhythm perception, with a participation of the visually deprived occipital brain areas whatever the sensory modality for input. We conclude that the visually deprived cortex switches its input modality from vision to audition and vibrotactile sense to perform this temporal processing task, supporting the concept of a metamodal, multisensory organization of this cortex.

## 1. Introduction

In adults affected by early visual deprivation, a crossmodal recruitment of the occipital cortex has been observed during auditory, tactile and olfactory processing [1,2,3,4,5,6,7] and during higher-level cognitive tasks such as verbal processing, complex haptic tasks, memory tasks or braille reading [8,9,10,11,12,13]. A link between this occipital cortex recruitment and improved perceptual and cognitive abilities has been evidenced, using functional magnetic resonance imaging (fMRI) and transcranial magnetic stimulation (TMS) in early blind humans, during sound localization [14,15], verbal memory tasks [8,16] or Braille reading [17,18].

Although most studies focused on the crossmodal changes in the visual brain areas in tasks performed by different sensory modalities, intramodal changes have also been observed in the auditory [19,20,21] and the somatosensory cortices [22,23,24,25] indicating that early blindness also trigger changes in brain areas dedicated to the preserved sensory modalities. Moreover, in blind subjects functional connectivity reorganizations between the sensory and heteromodal brain areas have been observed [9,26], highlighting that neuroplastic modifications may also arise in brain networks larger than the visual cortices, including regions associated with multimodal integration [27,28]. These integration areas have been described as critical for higher cognitive functions involved in the construction of an integrated, multisensory experience [29,30].

Beat perception is a multisensory experience that requires the ability to extract higher-order features of temporal information [31,32,33]. It remains under-investigated in early blind individuals. Beat perception can be achieved via most sensory modalities as shown in behavioral and neuroimaging studies, although it is best accomplished in audition [31,34,35,36,37,38]. We have previously shown that early blind individuals had finer beat asynchrony detection and outperformed sighted controls matched for musical experience [39]. These behavioral differences could be related to an improvement in beat extraction by the blind or with a more accurate mental representation of beat, putatively associated with changes in brain areas involved in beat processing.

Beat processing recruits a cortico-subcortical network that includes both sensory and heteromodal brain areas, such as the prefrontal cortex (PFC), the supplementary motor area (SMA), the premotor cortex (PMC) and the basal ganglia (BG) [31,37,40,41,42,43,44,45,46]. Previous reports let consider the basal ganglia as a key-role structure in the perception and production of a beat in the auditory modality [35,37,41,47]. We previously demonstrated that a common neural network was recruited during a beat detection task using auditory, visual or vibrotactile input [48]. This network included the prefrontal cortex (PFC), the supplementary motor area (SMA), the premotor cortex (PMC), the right inferior parietal lobe (IPL), the posterior superior temporal gyrus (pSTG), the anterior insula, the cerebellum and the putamen, indicating a convergence of perceptual information from different sensory modalities into a centralized representation of the beat.

Here, we used fMRI to highlight the neural network of beat detection in the auditory and vibrotactile modality in early blind individuals compared to sighted controls. We performed a rhythm discrimination task (using sequences with and without a beat) instead of sensorimotor synchronization in order to avoid brain activity generated by movement. We hypothesized that functional brain changes in the neural network of beat detection would be present in early blind (EB) individuals when compared to sighted controls, given the behavioral differences previously observed [39]. In other words, we hypothesized that EB subjects would outperform control subjects in detecting beats, and that certain plastic changes in their brain organization would be associated with these better skills. We wished to examine to what extent these changes involved the visually deprived occipital cortex (i.e., crossmodal brain plasticity), despite the non visual nature of the task (temporal extraction of a beat), and which heteromodal regions were also recruited among those known to be dedicated to auditory and vibrotactile beat perception (i.e., plasticity in brain areas outside sensory cortices). We hypothesized that the occipital cortex of blind people would be activated independently of the sensory input, according to the supramodal, multisensory organization hypothesis [29], but with a different recruitment between the sequences with and without a beat because usually the recruitment of visually deprived areas is task-dependent [6,49].

## 2. Materials and Methods

### 2.1. Subjects

The study involved 12 early blind individuals (EB) and 12 sighted controls (SC), matched for gender (9 men), age (EB: 38.3 ± 11.5; SC: 35.8 ± 14.6, *p* = 0.66) and the number of years of formal musical lessons (EB: 6.9 ± 5.0; SC: 6.8 ± 4.4, *p* = 0.93). The present study involved the same EB subjects as in a previous study [39] and musical experience was taken into account in order to control for its potential effect on beat and temporal perception [50,51,52]. The EB subjects were affected by complete blindness (no residual vision) as the result of bilateral ocular or optic nerve lesions from birth or before the age of 18 months. A summary of their medical history is provided in Table 1. All blind participants were proficient braille readers with two hours practice daily. They all learned it around the age of 5 years, before entering primary school. Sighted participants were studied blindfolded. No participant reported neurological or psychiatric illness. Two blind participants had a hearing aid and reported to hear normally with it. Written informed consent was obtained before the experiment, the protocol of which had been approved by the Biomedical Ethics Committee of the school of Medicine of the Université catholique de Louvain.

### 2.2. Equipment and Stimuli

Auditory stimuli were created using Adobe® Audition CS6. They consisted in stimulation sequences of 500 Hz pure tones of variable durations played in stereo (±75 dB) and delivered through an MRI compatible sound delivery system (NordicNeuroLab®, Bergen, Norway). Every subject confirmed he could hear the stimuli. The magnet noise was attenuated by headphones (30–35 dB) and earplugs (about 30 dB). Vibrotactile stimuli were 100 Hz pure tones of variable durations created and played using Labview®, Austin, TX, USA software and delivered using an MRI compatible tactile stimulator (sound card: SoundBox 7.1 Conrad®, Mortsel, Belgium) with ceramic piezoelectric bending elements (i.e., benders Q220-A4-303YB, Piezo Systems, Quick Mount Bender®, Woburn, MA, USA) applied on the index finger of both hands.

Two types of rhythmic stimulation sequences were used in each sensory modality as described [48]: 12 “beat” sequences and 12 “no beat” sequences were adapted from those used by Grahn and Rowe (2009, [37]). Briefly, “beat” sequences consisted of a train of sound/tactile stimuli with inter-onset-intervals (IOIs) with integer ratio relationships (e.g., 250, 500 and 1000 ms IOIs constituted a 1:2:4 relationship between onsets). Filled interval sequences, which increase stimulus saliency relative to silent interval sequences [37,53], were created with inter-beat-intervals (IBIs) ranging from 220 to 270 ms in 10 ms increments, and a fixed silent gap of 80 ms between stimuli to induce different tempi. The perception of regular temporal organization led to the perception of the beat [35,54]. No beat stimuli were created for this same range of IBIs (220–270 ms) and consisted of stimulus trains in which IOIs were randomly drawn from a set of ratios equal to 2/3×, 1× and 4/3× the IBI. An example of a auditory “beat” and “no beat” sequence is provided in Appendix A.

### 2.3. Behavioral Procedure

The week preceding the fMRI experience, each subject underwent a familiarization that consisted in presenting a “beat” and a “no beat” sequence in each modality. Then, subjects performed a behavioral test consisting in a “beat vs. no beat” detection task on ten rhythmic sequences including five “beat” sequences in each modality: audition (A) and tact (T). The presentation order of the modality (i.e., AT and TA) and sequences were pseudo-randomized and counterbalanced across subjects. Participants delivered their responses by pressing the right button of a mouse (Razer ®, model number: RZ01-0015) held in the left hand. Subjects were required to determine as quickly as possible whether a beat was present in the stimulation sequence. A feedback was provided after each response and the response times were recorded. Behavioral data (accuracy and response time) were analyzed using STATA/SE 12.0 for Windows ® (StataCorp, College Station, TX, USA). Response times related to incorrect responses were excluded from analysis.

### 2.4. 3D-MRI and fMRI Acquisition

During the fMRI session, subjects were instructed to determine whether a beat was present in each sequence. In the magnet, no motor response was required from the subject to keep at a minimum the motor-related brain activity. We used a block design paradigm with twelve experimental blocks (18 s per active epoch of stimulation condition, 8 brain volume repetitions) alternating with resting state periods (13.5 s, 6 brain volume repetitions) in four runs of 391.5 s each (2 runs for each modality). In each run, six “beat” and six “no beat” sequences alternated randomly. The sensory modality was briefly announced via headphones during the preceding resting period. Sighted subjects were studied blindfolded. Structural brain imaging was obtained for all subjects in the bicommissural (AC-PC) orientation [55] on a 3 Tesla MRI unit (Achieva, Philips Healthcare^®^, Best, The Netherlands) using a 3D fast T1-weighted gradient echo sequence with an inversion prepulse (Turbo field echo (TFE), TR (repetition time) = 9 ms, TE (echo time) = 4.6 ms, flip angle = 85 degree, 150 slices, 1 mm thickness, in plane resolution = 0.764 mm × 0.764 mm). The field of view was 220 mm × 197 mm, and the SENSE factor (parallel imaging) was 1.5. We used a thirty-two channels phased array head coil. Blood oxygen level dependent (BOLD) fMRI data were acquired using a 2D single shot T2*-weighted gradient echo-planar imaging (EPI) sequence (TR = 2250 ms, TE = 27 ms) with 41 axial slices (thickness = 3 mm), in the AC-PC orientation. The field of view was 220 mm^2^. The in-plane resolution was 2.75 mm^2^.

### 2.5. fMRI Data Analysis

Data analysis was performed using BrainVoyager QX 2.8 software package (Brain Innovation™, Maastricht, The Netherlands) with standard preprocessing procedures. Functional MRI data preprocessing included slice scan time correction, head motion correction and high-pass filtering (cutoff frequency: 2 cycles/run) in the frequency domain. Functional and anatomical data sets for each subject were coregistered and the resulting matching brain images were fit to the standardized Talairach space [55] and resliced (voxel size: 3 mm × 3 mm × 3 mm). Single-subject functional data were spatially smoothed (5 mm FWHM) in order to reduce intersubject anatomical variability. Functional data were further analyzed using a multiple regression model (general linear model (GLM), [56]) that consisted in the four experimental conditions (“beat” and “no beat” in each of the two modalities). In this model, the beta weights quantified the potential contribution of each predictor in each voxel time course. The predictor time courses of the regression model were computed on the basis of a linear model of the relation between neural activity and hemodynamic response, assuming a rectangular neural answer convolved with a standard hemodynamic response function (HRF) during phases of active conditions [56,57,58]. A random-effects (RFX) group analysis [59] was performed at the whole-brain level. Given our a priori hypotheses, all the analyses were performed using a threshold of *p* < 0.005 in combination with a cluster size threshold adjustment (minimum cluster size: 189 voxels) to achieve a corrected *p* < 0.05. This was performed based on the Monte Carlo simulation approach, extended to 3D data sets using the threshold size plug-in Brain Voyager QX [60]. Analysis of fMRI data was performed using the general linear model (GLM, two-way ANOVA with F tests), with the condition (“beat” vs. “no beat”) as a “within-subjects” factor and the group (early blind (EB) vs. sighted controls (SC)) as a “between-subjects” factor in each modality considered separately. This analysis was used to assess the condition (“beat vs. no beat”) by group (“EB vs. SC”) interaction effect, the group effect and the condition effect. To measure the link between beat detection performance (response time in behavioral study) and brain activity (fMRI beta values) in each subject group, a RFX covariate analysis was performed (linear regression) at the whole-brain level to examine whether the activation in the (“beat minus no beat”) contrast could be explained by the influence of individual performance, using the individual response time for beat detection in the corresponding sensory modality as a covariate.

## 3. Results

### 3.1. Behavioral Results

A 2 (condition: “beat vs no beat”) × 2 (group: early blind (EB) vs. sighted control (SC) subjects) two-way analysis of variance (ANOVA with F tests) was performed on the response times for “beat/no beat” discrimination in each modality considered separately. We observed no effect of the condition (F(1.42) = 0.05, *p* = 0.829), no effect of the group (F(1.42) = 0.15, *p* = 0.704) and no interaction (F(1.42) = 0.03, *p* = 0.866) in the auditory modality. We observed no effect of the condition (F(1.42) = 0.17, *p* = 0.684), no effect of the group (F(1.42) = 0.01, *p* = 0.911) and no interaction (F(1.42) = 0.66, *p* = 0.422) in the vibrotactile modality. The mean response times for “beat/no beat” detection in auditory “beat” and “no beat” sequences were 6.32 ± 1.79 s and 6.60 ± 2.10 s in the SC group and 5.93 ± 2.93 s and 5.96 ± 2.97 s in the EB group. The mean response times for “beat/no beat” detection in vibrotactile “beat” and “no beat” sequences were 6.79 ± 2.46 s and 8.17 ± 3.05 s in the SC group and 7.44 ± 4.04 s and 7.03 ± 4.14 s in the EB group. There was no significant difference between audition and the vibrotactile modality in either group (all *p* < 0.05), although the response times were slightly shorter in the auditory modality. The accuracy (correct responses, from 0 to 10), averaged for all 10 items, was compared in each modality separately. A non parametric Mann–Whitney test performed on these scores revealed differences between early blind participants and their controls (audio accuracy for early blind participants: median score = 10.000 (25–75%: 9.250–10.000); audio accuracy for controls; median score = 9.000 (25–75%: 8.250–9.750, *p* = 0.011); tact accuracy for early blind participants: median score = 9.000 (25–75%: 8.250–10.000); tact accuracy for controls: median score = 8.000 (25–75%: 6.000–9.000, *p* = 0.040). A graphical representation or behavioral results is provided in the Appendix A.

### 3.2. fMRI Results from Whole Brain ANOVA

#### 3.2.1. Main Effect of the Condition: Brain Activation Specific to Beat Detection

To identify the neural network that specifically supported beat detection in each modality, we contrasted the “beat” with the “no beat” condition in audition and in the vibrotactile modality, separately. In audition, the main condition effect in GLM, i.e., the contrast (“beat” minus “no beat”) revealed bilateral brain activity in the putamen, the inferior parietal lobule (IPL, Brodmann area (BA) 40) and the medial frontal gyrus (MeFG) or supplementary motor area (SMA, BA 6). There were also activation foci in the left dorsal premotor cortex (PMC, BA 6) and in the left superior temporal gyrus (STG, BA 22), and the left middle (MiFG, BA 9) and inferior frontal gyrus (IFG)/ventral PMC (BA 44/BA 6) (Figure 1, Table 2). No activation focus was found in the reverse contrast (“no beat” minus “beat”).

In the vibrotactile modality, the F contrast (“beat” minus “no beat”) showed bilateral brain activity in the putamen. There were also activation foci in the right inferior parietal lobule (IPL, BA 40), in the right middle frontal gyrus (MiFG, BA 9) and in the left insula and inferior frontal gyrus (IFG)/ventral PMC (BA 44/BA 6) (Table 3, Figure 1). No activation focus was found in the reverse contrast (“no beat” minus “beat”). These brain areas are part of the supramodal neural network involved in the processing of a beat detected via different sensory modalities [48].

#### 3.2.2. Main Effect of the Group: Differential Brain Activation in EB and Controls during Beat Processing

To identify the neural network specifically activated in blind participants compared to controls, we contrasted the group activation maps in audition and in the vibrotactile modality, separately. In audition, the main group effect in GLM, i.e., the comparison (early blind (EB) minus sighted controls (SC)) revealed brain activity in the right superior temporal gyrus (STG, BA 22), insula (BA 13) and inferior frontal gyrus (IFG, BA 46), and in the left hippocampus and parahippocampal gyrus (Hi, BA30) and inferior parietal lobule (IPL, BA 40). In addition to these brain areas, brain activation foci were observed in visual association areas, in and around the left inferior temporal and fusiform gyrus (ITG/FG, BA 37). There was also a trend to higher activation in the left middle occipital gyrus (MOG, BA 19) in EB subjects (Table 2, Figure 2). No activation focus was found in the reverse contrast (SC minus EB). In the vibrotactile modality, the F contrast (EB minus SC) showed bilateral brain activity in the hippocampus and parahippocampal gyrus (Hi, BA35/30) and in the cuneus (Cu, BA 18) and in other visual association areas including the right precuneus (PCu, BA 7) and the left lingual gyrus (LG, BA 19) (Table 3, Figure 2). No activation focus was found in the reverse contrast (SC minus EB).

#### 3.2.3. Interaction: Brain Activation Specific to Beat Detection in EB Subjects Compared to Controls

In the auditory modality, the group (EB vs. SC) × condition (“beat” vs. “no beat”) interaction in GLM revealed bilateral recruitment of the lingual gyrus (LG, BA 18), with a trend to activate other visual association areas including in and around the right inferior temporal and the middle occipital gyrus (ITG/MOG, BA 37/19) bilaterally (Table 2, Figure 3). In the vibrotactile modality, the group (EB vs. SC) × condition (“beat” vs. “no beat”) interaction in the GLM revealed bilateral recruitment of frontal brain areas including the precentral, inferior and middle frontal gyrus (PrCG/IFG/MFG, BA 6/44/46), the medial frontal gyrus (MeFG) or supplementary motor area (BA 6) and the insula (BA 13). An additional brain activation focus was observed in the right cingulate gyrus (BA 24). There was also a trend to brain activation in and around the right middle occipital gyrus (BA 37/19) (Table 3, Figure 3).

#### 3.2.4. Covariance Analysis: Relationship between Response Times and Brain Activity

In each group, the activation map resulting from the analysis of covariance between the brain activity for the contrast (“beat” minus “no beat”) and the individual response time for beat detection in the corresponding modality (used as covariate) is shown in Figure 4. As previously observed during a beat detection task using auditory, visual or vibrotactile input [48], there was a significant (negative) correlation in the putamen in both groups at the selected threshold (*p* < 0.005) (Table 4, Figure 4). Several other brain areas showed a significant (either negative or positive) covariation at the selected threshold in both groups, in particular the middle and inferior frontal gyrus (BA 9/46), precentral/middle frontal gyrus (BA 6) and inferior parietal lobule (BA 40) in the right hemisphere and the left insula (BA 13). Despite the absence of behavioral differences between groups, the brain activity maps from the analysis of covariance differed between the two groups: in EB subjects, there were additional brain areas that covariate with individual response times, positively in the right middle and inferior temporal gyrus (BA 21, BA 37/19), the right insula (BA 13) and in the right superior, middle and inferior occipital gyrus (BA 19/37), lingual gyrus (BA 19) and cuneus (BA 17/18), negatively in the left superior and middle frontal gyrus (BA 6/46), the left superior temporal gyrus (BA 22) and the left cuneus (BA 18).

## 4. Discussion

To our knowledge, this is the first fMRI study to compare the brain activity involved in beat detection in early blind (EB) and sighted control (SC) subjects using both audition and vibrotactile stimuli for input. Contrary to our hypothesis, the EB subjects did not outperform the controls in this task: in both groups, response times were similar and both groups performed at the ceiling, although accuracy was slightly better in EB than in SC. Despite this similar performance in the behavioral study, there were differences in brain activation between EB and SC during fMRI that involved, in both modalities, sensory brain areas, such as occipital regions, and heteromodal regions, notably parietal and prefrontal cortical areas. The covariance analysis provided evidence of a correlation between brain activation and behavioral performance in several prefrontal and visual brain areas in EB subjects only.

### 4.1. Specific Neural Network for Beat Detection

In both modalities, when comparing “beat” with “no beat” conditions, we observed the activation of prefrontal brain regions, dorsal premotor areas, associative regions (i.e., the inferior parietal lobule, BA 40) and the basal ganglia, notably the putamen. The recruitment of these regions is in line with the neural network previously described for beat detection mainly in audition [31,47,61,62]. It is also part of the neural network of beat detection we observed in the auditory, visual and vibrotactile modalities in a previous study carried out in sighted participants [48]. There were several previous indications that interactions of premotor and prefrontal areas with basal ganglia were involved in the analysis of temporal sequences and in the prediction of a beat [37,44,47,61,63,64]. The inferior parietal lobule is considered as a multimodal association area that links perception and action, acting as an interface between sensory areas and motor planning areas, matching the information coming from the first with the information generated by the second [65]. Within this supramodal network for beat processing, the putamen has constantly showed the highest specificity to the beat in all modalities, as in the present study. The role of basal ganglia in the perception of rhythm is the prediction of the beat, i.e., a detectable periodicity in the temporal structure [37]. At a neuronal level, the presence of the beat has been related with an increase in the putamen activation [31,37,66].

### 4.2. Differences between Early Blind and Controls in Beat Processing

In the present study, there was crossmodal activation of the occipital brain areas in EB subjects only, while they perceived rhythmic auditory and vibrotactile stimulation. This is in line with several previous observations of a crossmodal recruitment of occipital brain regions in early blind subjects during physiological tasks involving auditory, tactile or olfactory stimulation [3,5,6]. There are several reports of a recruitment of occipital brain regions in the blind when performing spatial tasks in auditory and vibrotactile modalities [6] leading to consider that the visual system is particularly efficient in spatial processing [67]. Here we observed that a temporal processing task, involving the perception of rhythm, activates the visually deprived cortex, although temporal processing is generally processed more precisely by the auditory system than the visual due to higher temporal resolution favoring audition [68,69,70,71]. The present results indicate the possible recruitment of occipital brain regions by the blind in either temporal or spatial tasks. The occipital brain areas that covariate with performance in blind subjects are similar to occipital regions activated above rest during a beat detection task in the visual modality in sighted subjects [48]. This cross-modal plasticity of the occipital brain areas associated with temporal processing in the blind could be due to an expansion of the multisensory properties of these brain regions, in accordance with a metamodal organization of the brain [29,72], which could be useful for compensatory brain plasticity after early sensory deprivation. An additional support to this view is the recent observation of auditory cortex activation in deaf subjects during a rhythm discrimination task performed in vision [73].

When comparing the brain activation patterns in EB and CS, we also observed differences in heteromodal brain regions, notably in the prefrontal and parietal cortex. These differences could further indicate a reorganization of functional connectivity between sensory and heteromodal brain areas as previously depicted in blind subjects [9,26]. Interestingly, some authors have proposed that beat perception involves a transformation from perception to action [33] and in this hypothesis, sensory regions and motor planning areas are connected via the parietal cortex to anticipate movement related to rhythm. It should be noted that, in subjects with congenital or late blindness, there are previous reports of interconnectivity increases between multimodal integration cortices and the auditory, sensory-motor and visual systems [28]. All of this evidence points to the concept that brain plasticity induced by early blindness can also emerge from cortical networks beyond visual areas, including brain regions associated with multimodal integration [28] that play a key role in building an integrated multisensory experience [30]. The integrated implication of both heteromodal and sensory brain areas in beat detection is not completely unexpected since the processing of the metric structure of rhythm has similarities with the syntax structure of language, which is also processed at a high cognitive level [74,75,76].

### 4.3. Differences between EB and SC Subjects in Beat (Minus No Beat) Detection

Interestingly, there were differences in the occipital recruitment in the blind participants between the “beat” and “no beat” conditions, which indicates further that the recruitment of visually deprived brain areas is task-dependent or at least modulated by the task. The increased recruitment of heteromodal frontal brain areas during beat detection by the blind, especially in the vibrotactile modality, is more puzzling. It could be the neural substrates needed in complex tasks involving beat detection abilities, since early blind individuals had a lower beat asynchrony detection threshold than matched sighted controls in a previous study [39] and an equal good performance compared to SC subjects in the present study. As a result, EB might possess similar capabilities of extracting and synchronizing beats as sighted controls and/or accurate mental representation of beats despite the lack of vision, provided there is a higher recruitment of heteromodal brain areas. Interestingly, in a study investigating the neural correlates of interindividual differences in beat perception, increased activation was observed in heteromodal brain regions, notably in the left frontal operculum and the left supplementary motor area, in sighted individuals who could extract a more abstract beat from a rhythm sequence (i.e., a beat that was not based on each sound of the sequence) [77].

### 4.4. Limitations of the Study

Most of the participants in both groups were trained musicians, skilled to easily perform tasks similar to the beat/no beat detection task used in the present study. This could prevent seeing behavioral differences between groups. A single temporal processing task (beat/no beat detection) was tested in the present study. Although there were differences in occipital recruitment in blind participants between the “beat” and “no beat” conditions, indicating that recruitment of visually deprived brain areas was task-modulated, the study does not provide a definite answer to the question of the nature of the activation of the occipital cortex (task-driven, metamodal or multisensory, pluripotent and potentially recruited in any task [78]) in early blind subjects. Additional studies with different temporal processing tasks are clearly needed to disentangle the two hypotheses and to further elucidate the origins of recruitment of specific subregions among visual brain areas that have been activated in blind subjects.

## 5. Conclusions

Compared to blindfolded controls, we observed that beat detection by blind people using auditory or vibrotactile stimulation recruited a large brain network, with higher brain activity in the visually deprived cortex and in heteromodal regions, especially the parietal and prefrontal cortex. These changes in brain activity support the concept of a supramodal network involved in beat detection by the blind, similar to previous observations in sighted subjects, but associated with crossmodal recruitment of occipital brain areas. Although more studies are needed to determine why heteromodal brain areas were more recruited in the vibrotactile modality, the study is consistent with the model that information travels from the primary sensory cortex to multimodal regions to reach associative brain areas that support cognitive functions in all subjects, but with a shift from the unimodal sensory cortex in sighted subjects to the visually deprived cortex in blind subjects.

## Figures and Tables

**Figure 1 life-11-00296-f001:**
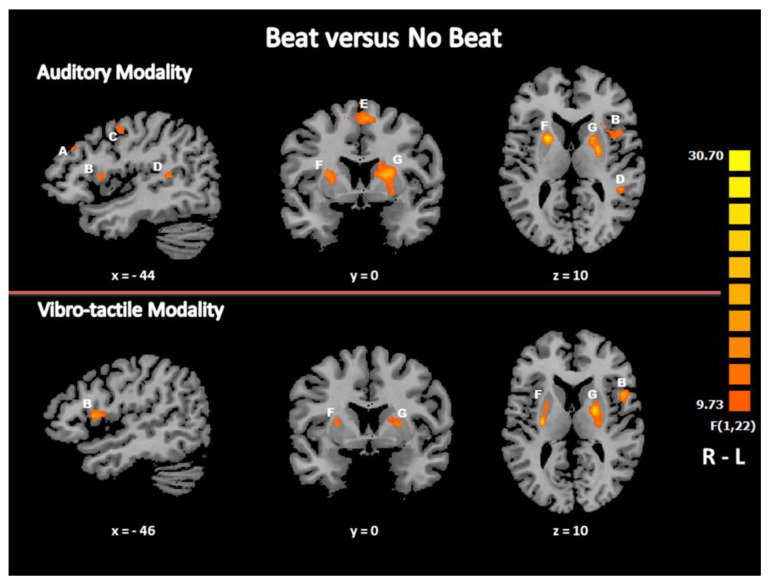
Brain activation maps specific to beat detection. Regions activated in each modality in the contrast (“beat” minus “no beat”) are displayed according to the color scale that codes for the activation level based on the t-values. A: Middle frontal gyrus (BA 9); B: inferior frontal gyrus (BA 44/6); C: precentral gyrus/middle frontal gyrus (BA 6); D: superior temporal gyrus (BA 22); E: medial frontal gyrus (BA 6); F - G: putamen; R: right; L: left.

**Figure 2 life-11-00296-f002:**
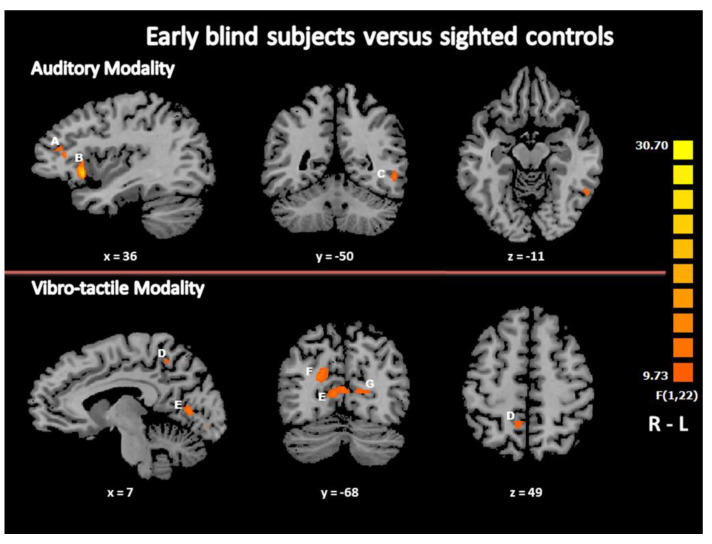
Differences in brain activation during beat processing as a function of the group. Regions activated in each modality in the contrast (early blind (EB) minus sighted controls (SC)) are displayed according to the color scale that codes for the activation level based on the t-values. A: Inferior frontal gyrus (BA 46); B: insula (BA 13); C: inferior temporal gyrus/fusiform gyrus (BA 37); D: precuneus (BA 7); E - F - G: cuneus (BA 18); R: right; L: left.

**Figure 3 life-11-00296-f003:**
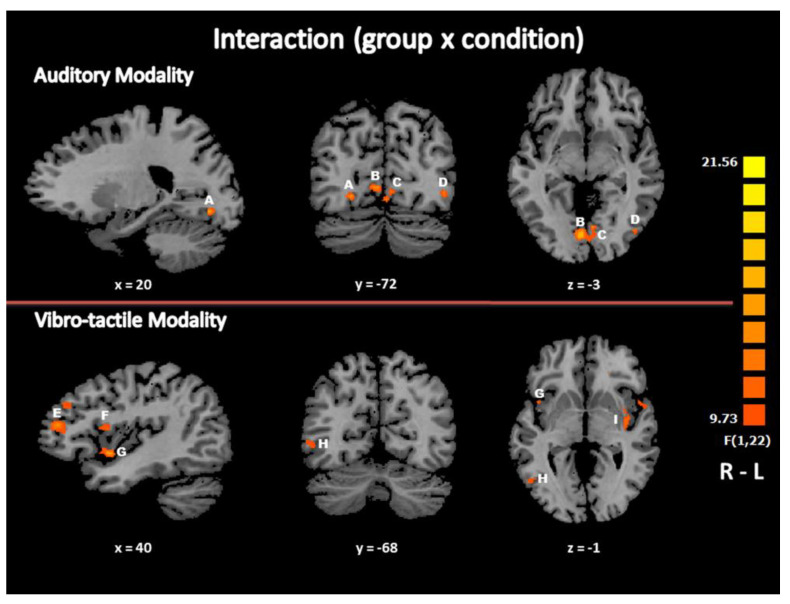
Differences in brain activation specific to beat detection as a function of the group. Brain areas with a higher activity level in blind participants compared to controls and specific to beat detection in each modality (group by condition interaction maps in GLM) are displayed according to the color scale that codes for the activation level based on the t-values. A - B - C: Lingual gyrus (BA 18); D: inferior occipital gyrus (BA 19); E: middle frontal gyrus (BA 46); F: inferior frontal gyrus (BA 44); G: insula (BA 13); H: middle temporal gyrus/middle occipital gyrus (BA 37/19); I: insula (BA 13); R: right; L: left.

**Figure 4 life-11-00296-f004:**
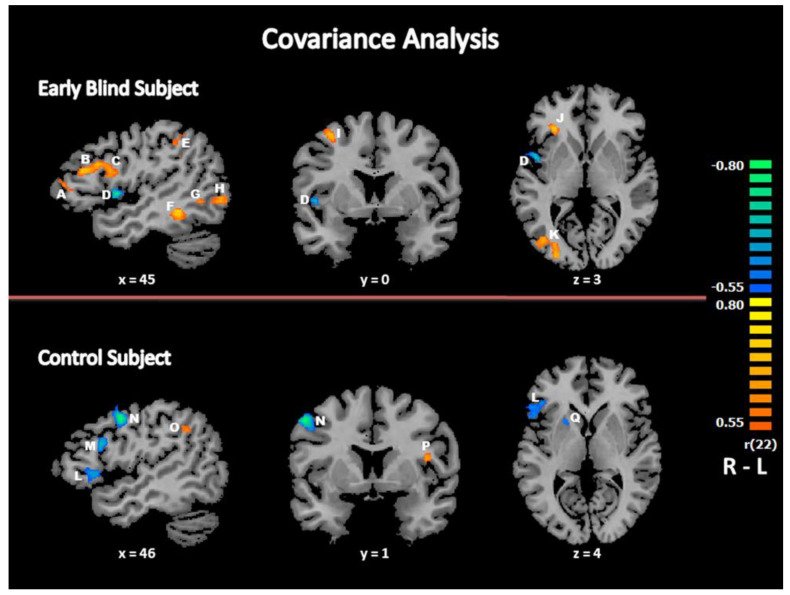
Relationship between beat detection performance and brain activity. The activation maps were obtained using an analysis of covariance between the brain activity for the contrast (“beat” minus “no beat”) and the individual response time for beat detection in the corresponding modality used as the covariate (*p* < 0.005). The color scale codes for the activation level based on the *r* coefficient. A: Inferior frontal gyrus (BA 46); B: middle frontal gyrus (BA 46); C: inferior frontal gyrus (BA 9); D: insula (BA 13); E: inferior parietal lobule (BA 40); F: fusiform gyrus (BA 37); G: inferior temporal gyrus/middle occipital gyrus (BA 37/19); H: inferior occipital gyrus (BA 19); I: precentral gyrus/middle frontal gyrus (BA 6); J: inferior frontal gyrus (BA 45/46); K: middle occipital gyrus (BA 19); L: inferior frontal gyrus (BA 45/46); M: middle frontal gyrus (BA 9); N: middle frontal gyrus/precentral gyrus (BA 6); O: inferior parietal lobule (BA 40); P: insula (BA 13); Q: putamen. R: right; L: left.

**Table 1 life-11-00296-t001:** Profile of the blind subjects.

Subjects	Age	Sex	Handedness (**)	Onset of Blindness	Diagnosis	Musical Experience (***)
EB 1	37	M	R	Birth	Leber congenital amaurosis	7
EB 2	40	F	A	Birth	Retinopathy of prematurity	7°
EB 3	29	M	R	Birth	Persistent hyperplastic primary vitreous involving both eyes	6
EB 4	57	M	R	Birth	Severe retinal dystrophy (*)	13
EB 5	46	M	R	Birth	Leber congenital amaurosis	3
EB 6	37	M	R	Birth	Anterior chamber cleavage syndrome (Peters syndrome)	14°
EB 7	32	F	R	Birth	Severe corneal dystrophy (*)	1°
EB 8	62	M	R	<18 months	Bilateral retinoblastoma	15
EB 9	28	M	L	Birth	Hereditary retinal dysplasia (*)	7
EB 10	35	F	R	Birth	Retinopathy of prematurity	2
EB 11	24	M	R	Birth	Severe optic nerve dystrophy (*)	none
EB 12	32	M	R	Birth	Genetic eye disorder (*)	8°

Note: EB: early blind; all subjects were totally blind. As subject EB8 had very poor vision from birth and underwent a bilateral eye enucleation by the age of 18 months, he was considered early blind. He did not remember any visual experience. (*): no additional details available. (**) The handedness was verbally reported by the subjects. R: right, L: left, A: ambidextrous. (***) Musical experience: number of years of formal musical lessons. ° Subjects who had some percussion experience.

**Table 2 life-11-00296-t002:** Brain activation foci from two samples F-test in the auditory modality.

F-Test (within) “Beat” Minus “No Beat”							
Brain region	Brodmann area	Coordinates			Cluster size	F-value	*p*-value
		x	y	z			
R Inferior Parietal Lobule	BA 40	57	−29	22	262	18.0404	0.000330
R_Cerebellum		27	−55	−23	1384	38.9461	0.000003
R Putamen		21	5	10	1113	31.3233	0.000013
R/L Medial Frontal Gyrus	BA 6	−9	−13	55	4156	38.9245	0.000003
L Cingulate Gyrus	BA 24/32	−12	14	34	314	24.8991	0.000054
L_Putamen		−27	−4	16	5197	33.9139	0.000007
L Inferior Parietal Lobule	BA 40	−30	−52	40	405	17.0245	0.000443
L Middle Frontal Gyrus	BA 9	−39	35	31	1363	24.7331	0.000056
L Precentral /Middle Frontal Gyrus	BA 6	−48	−7	46	377	18.9000	0.000258
L Inferior frontal/Middle Frontal Gyrus	BA 44/6	−41	8	10	277	24.2598	0.000063
L Superior Temporal Gyrus	BA 22	−48	−43	19	370	15.2869	0.000751
**F-test (between) EB Minus SC**							
Brain region	Brodmann area	Coordinates			Cluster size	F-value	*p*-value
		x	y	z			
R Superior Temporal Gyrus	BA 22	57	−10	4	249	15.8472	0.000632
R Insula	BA 13	36	17	−2	719	21.9237	0.000114
R Inferior Frontal Gyrus	BA 46	36	32	13	207	14.8069	0.000573
L Parahippocampal Gyrus/Hippocampus	BA 30	−6	−40	1	256	17.8974	0.000344
L Inferior Parietal Lobule	BA 40	−42	−37	37	197	18.0088	0.000333
L Inferior Temporal Gyrus/Fusiform Gyrus	BA 20/37	−51	−49	−5	414	13.2791	0.000431
L Middle Occipital Gyrus (*)	BA 19	−52	−67	−2	102	11.8650	0.000612
**F-Test (Interaction) within “Beat” vs “No Beat” and between EB vs SC**							
Brain region	Brodmann area	Coordinates			Cluster size	F-value	*p*-value
		x	y	z			
R Lingual Gyrus	BA 18	21	−73	−8	249	19.8387	0.000199
R Lingual Gyrus	BA 18	6	−79	−5	899	20.5595	0.000164
R Inferior Temporal Gyrus/Middle Occipital Gyrus (*)	BA 37/19	42	−64	−5	84	12.4729	0.002651
L Lingual/Fusiform Gyrus	BA 18/19	−3	−70	−11	677	16.5092	0.000517
L Middle Temporal Gyrus /Middle Occipital Gyrus (*)	BA 37/19	−42	−70	7	123	14.2863	0.001031
L Middle Occipital Gyrus (*)	BA 19	−42	−73	−5	123	16.3655	0.000540

(*) shown here for reference, though below the cluster size threshold; (“Beat” minus “No Beat” and EB minus SC), ANOVA, *p* < 0.005.

**Table 3 life-11-00296-t003:** Brain activation foci from two samples F-test in the vibro-tactile modality.

**F-Test (within) “Beat” Minus “No Beat”**								
Brain region	Brodmann area	Coordinates			Cluster size		F-value	*p*-value
		x	y	z				
R Inferior Parietal Lobule	BA 40	28	−46	55	213		17.6941	0.000364
R Putamen		27	−16	10	537		29.3930	0.000019
R Middle Frontal Gyrus	BA 9	24	38	22	300		15.1124	0.000793
L Putamen		−21	−7	12	1480		30.7040	0.000014
L Insula	BA 13	−36	−25	25	739		30.5384	0.000015
L Precentral Gyrus/Inferior Frontal Gyrus	BA 6/44	−48	11	10	699		25.8972	0.000042
**F-Test (between) EB Minus SC**								
Brain region	Brodmann area	Coordinates				Cluster size	F-value	*p*-value
		x	y	z				
R Parahippocampal gyrus/Hippocampus	BA 35/30	15	−40	−8		975	22.3237	0.000103
R Cuneus	BA 18	21	−67	19		682	17.1111	0.000432
R Precuneus	BA 7	6	−49	49		249	14.4397	0.000981
L Cuneus	BA 17/18	−2	−75	7		1073	17.0310	0.000443
L Lingual Gyrus	BA 19	−27	−58	0		498	17.7397	0.000360
L Parahippocampal gyrus/Hippocampus	BA 35/30	−27	−40	−8		989	22.8442	0.000090
**F-Test (Interaction) within “Beat” vs “No Beat” and between EB vs SC**								
Brain region	Brodmann area	Coordinates				Cluster size	F-value	*p*-value
		x	y	z				
R Precentral Gyrus /Inferior Frontal Gyrus	BA 6/44	57	8	13		2208	23.1492	0.000083
R Middle Frontal Gyrus	BA 46	39	41	16		979	21.4826	0.000128
R Insula	BA 13	39	2	−5		243	27.3261	0.000030
R Insula	BA 13	27	20	13		529	28.1927	0.000025
R Cingulate Gyrus	BA 24	12	−7	43		214	14.0374	0.001117
R Medial Frontal Gyrus	BA 6	3	−7	64		234	13.8985	0.001168
R Middle Temporal Gyrus/ Middle Occipital Gyrus (*)	BA 37/19	51	−64	0		121	13.6849	0.001252
L Medial Frontal Gyrus	BA 6/8	−9	20	49		235	17.6665	0.000367
L Insula	BA 13	−39	−7	−2		473	17.4133	0.000396
L Middle Frontal Gyrus	BA 46	−49	29	22		2838	28.3219	0.000024
L Inferior Frontal Gyrus/Precentral Gyrus	BA 44/6	−48	7	7		1082	21.6273	0.000123

(*) shown here for reference, though below the cluster size threshold; (“Beat” minus “No Beat” and EB minus SC), ANOVA, *p* < 0.005.

**Table 4 life-11-00296-t004:** Brain activation foci in covariance analysis.

“Beat” Minus “No Beat”, Early Blind Subjects							
Brain region	Brodmann area	Coordinates			Cluster size	R	*p*-value
		x	y	z			
R Middle Frontal Gyrus	BA 9/46	52	26	25	549	0.7585	0.000017
R Inferior Frontal Gyrus	BA 44	54	11	25	1777	0.7548	0.000020
R Inferior Parietal Lobule	BA 40	36	−46	46	2477	0.7501	0.000024
R Superior Occipital Gyrus /Precuneus	BA 19/7	27	−67	31	1290	0.8035	0.000002
R Insula	BA 13	45	2	4	414	−0.7936	0.000004
R Middle Temporal Gyrus	BA 21	42	−40	−8	549	0.7613	0.000016
R Inferior Temporal Gyrus/Middle Occipital Gyrus	BA 37/19	36	−58	−5	379	0.7276	0.000056
R Lingual Gyrus	BA 19	30	−52	−5	376	0.7364	0.000041
R Inferior Occipital Gyrus/ Inferior Temporal Gyrus	BA 18/19	39	−76	−2	1627	0.7854	0.000005
R Inferior Frontal Gyrus	BA 45/46	27	29	7	514	0.8045	0.000002
R Precentral/Middle Frontal Gyrus	BA 6	33	−1	52	600	0.7613	0.000016
R Cuneus	BA 17/18	18	−85	13	236	0.7449	0.000030
R Medial Frontal Gyrus	BA 10	6	62	12	597	−0.7508	0.000024
L Superior Frontal Gyrus	BA 6	−9	17	52	1093	−0.8134	0.000001
L Cuneus	BA 18	−15	−73	25	592	−0.7620	0.000015
L Cuneus	BA 17/18	−18	−70	13	429	−0.7578	0.000018
L Putamen		−18	5	7	200	−0.7038	0.000124
L Middle Frontal Gyrus	BA 46	−36	44	19	474	−0.7487	0.000026
L Insula	BA 13	−48	−19	22	238	0.8065	0.000002
L Superior Temporal Gyrus	BA 22	−60	−53	13	725	−0.8415	0.000000
**“Beat” Minus “No Beat”, Control Subjects**							
Brain region	Brodmann area	Coordinates			Cluster size	R	*p*-value
		x	y	z			
R Middle Frontal Gyrus	BA 9	48	11	25	1642	−0.7310	0.000050
R Inferior Frontal Gyrus	BA 45/46	54	14	6	953	−0.7040	0.000123
R Middle Frontal /Precentral Gyrus	BA 6	48	−1	43	874	−0.8603	0.000000
R Inferior Parietal Lobule	BA 40	48	−49	34	252	0.6797	0.000259
R Putamen		21	11	7	291	−0.6559	0.000502
L Tail of Caudate Nucleus		−21	−28	25	341	0.7355	0.000042
L Insula	BA 13	−39	2	16	265	0.6710	0.000332

Covariate: response times for beat detection in behavioral procedure, *p* < 0.005.

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
