# Peer review of "Beat Detection Recruits the Visual Cortex in Early Blind Subjects"

_life, 2021, doi:10.3390/life11040296_

Round 1
Reviewer 1 Report
This is a very interesting study examining the crossmodal, neural basis for auditory and vibrotactile beat detection. The participant groups were well matched (early blind and blindfolded), and the experimental design and analysis look sound. My suggestions are primarily in regards to the interpretation of the findings and the discussion. I think the paper will be suitable for publication in Life upon minor revisions.
The regions in the occipital cortex that had greater activity in the early blind participants than the sighted are interesting, and the the focus of the study (given the title). It would be interesting to have more consideration of the different specific regions that were active. Is there anything particularly informative about the specific sub-regions that might elucidate the mechanisms or origins of the recruitment of these areas?
Would the authors consider the activation patterns to be more in line with metamodal views (occipital activity is task-based due to the particular computational abilities of the area) or pluripotent views (Bedny's idea that it is not task-specific, but simply that any task would recruit it in the sensory deprived brain)? It is interesting that on line 298 it is noted that there is a metric structure to the rhythm. Might that require something like the the spatial computations afforded by retinotopic maps in the occipital lobe, and thus in a sense be spatial (metamodal)? Or is that metric more aligned to the tonotopy of auditory cortex, and just happens to invade the occipital lobe due to a lack of other sensory use (pluripotent)? Is there any area you might conclude is metamodally crucial for beat detection, such that it is consistent across sensory input (auditory and tactile), and unique to the EB versus blindfolded groups?
It is interesting that there was no behavioural difference between EB and blindfolded, but there were the neural differences. Was there anything different about your task here compared to the prior studies that found behavioural differences?
Reviewer 2 Report
This is an excellent paper presenting an fMRI demonstration in recruitment of visual areas to a temporal task. This is somewhat surprising, to me at least, and fascinating. The study design is good and clearly described, the conclusions are appropriate and the statistical analysis is as expected.
